# Protecting a Pumping Pipeline System from Low Pressure Transients by Using Air Pockets: A Case Study

**Rafael Bernardo Carmona-Paredes** **, Oscar Pozos-Estrada \*, Libia Georgina Carmona-Paredes, Alejandro Sánchez-Huerta, Eduardo Antonio Rodal-Canales and Germán Jorge Carmona-Paredes**

Instituto de Ingeniería, Department of Hydraulic Engineering, Universidad Nacional Autónoma de México, Cd. Universitaria, C.P. 04510 Mexico City, Mexico

\* Correspondence: opozose@iingen.unam.mx; Tel.: +52-55-5623-3600

**Abstract:** This paper presents a case study of an existing water pipeline with five pumping stations each equipped with five pumps. In order to study the pipeline behavior prior to putting the system into operation, several transient simulations for different scenarios were developed. Results revealed that the most serious situation occurred when a simultaneous failure of the five pumps occur at each station caused by power cut, producing negative pressure waves because the system for control of hydraulic transients of the pipeline was insufficient to suppress downsurge pressures, due to the moment of inertia of all the pumps being erroneously considered during the design stage. The necessity to start supplying water to the population led to attempt an unconventional form of protecting the line against low pressures. The solution was to operate two of the five pumps per plant, and permit air to enter through combination air valves located along the pipeline. Air entrained formed pockets that remained stationary at the air valves locations, acting as air cushions that absorbed the energy of transient pressure waves. Computational simulations were conducted considering that two pumps are in operation at each plant and suddenly these fail simultaneously caused by power failure. The program was verified by comparing the calculated results with those registered during field pressure measurements. It was noticed that the surge modelling results are in good agreement with the measured data; furthermore, these show the air pockets in combination with existing devices for transient control protect the system adequately, avoiding potential damage to the pipeline.

**Keywords:** air pocket; air valve; pumping pipeline; pump failure; fluid transients; devices for transient control

## 1. Introduction

For several decades now, pumping pipeline systems have been used for transporting water and wastewater all over the world. During operation, pipelines are subjected to different maneuvers such as programmed shutdown or start-up of pumps, an unexpected loss of power on the pumps, as well as rapid closing or opening of valves. In consequence, the aforementioned situations can cause fluid transients, which can generate large pressure surges that may produce the failure of pumps and devices, and even system fatigue or pipe ruptures [1]. Therefore, it is of paramount importance to consider an effective surge control for pumping pipelines from the design stage.

Pipeline projects are commonly divided in three stages: modeling, design, and construction [2]. During the hydraulic transient modeling, the surge problems are identified and alternatives are evaluated and recommended based on the achieved results. Likewise, at the design stage, there is a

relevant aspect that has to be considered regarding the pipe wall thickness, which has to withstand the full range of transient pressures that the pipeline is subjected during its entire lifetime. Finally, at the construction stage, problems may occur that have to be solved based on good engineering judgement.

The effect of maximum pressures during transient events in pipelines may be either small or catastrophic. Pressure surges may only cause cracks in an internal lining; sometimes, this damage is not detected at the time, resulting in leakage and in intensified corrosion that over months or years can considerably reduce the wall thickness and, when combined with repeated transients, may cause the pipeline to collapse. More severely, strong positive pressures may damage connections and flanges between pipe sections, as well as valves or any device for transient protection. In the most extreme case, the maximum pressures may destroy pipelines, tunnels, valves, or other components, causing considerable damage and sometimes loss of human life [3].

Regarding minimum pressures, these can be extremely destructive as the maximum pressures. For instance, the transient pressures in a pipeline may drop to vapor pressure causing cavitation and water-column separation following power failure to a pump or sudden rapid closure of a valve. If the pressure is reduced to the liquid vapor pressure, a vapor cavity forms, and the water column in the pipeline separates at particular points. For this reason, it has to be analyzed whether the pressures produced when the separated water columns rejoin are acceptable. Likewise, concerning the pipe wall material, whether the pipeline is prone to suffer pipe crushing should be studied, when the hydraulic grade line intersects the pipeline profile, producing a vacuum [4,5].

Therefore, based on the above statements, the possibility of water-column separation should be investigated during the design stage of a pumping pipeline and, if necessary, appropriate control surge devices have to be provided to prevent this phenomenon, such as air chamber, surge tank, one-way surge tank, flywheels, air-inlet valves, among others. Air chambers, surge tanks and one-way surge tanks are usually costly. Likewise, an increase in the inertia of the pump-motor by using a flywheel increases the space requirements and may require a separate starter for the motor, thus increasing the initial project costs. Caution must be exercised if air-inlet valves are provided because, once activated, air admitted into a pipeline has to be removed from the line prior to refilling since entrapped air may result in very high pressures [4].

Furthermore, in addition to the appropriate surge suppression devices to prevent water-column separation, whenever possible, the use of a pump with a large motor rotating moment of inertia should be studied, since it can help to control transients due to the rotating parts of the pump and motor continually moving water through the pump for a longer time as they slowly decelerate after switching off the power [6]. In contrast, when the pumping equipment has a limited inertia of rotating parts, the pump speed will decrease very rapidly, generating low pressure waves, which could lead to water-column separation [4].

Likewise, another way to reduce the pressure transients in pipelines is by using large air pockets, provided that they remain stationary in an adequate location, normally at summits of the line. An air pocket can accumulate at a high point of a pipeline, when the drag force of water flow is not enough to remove it and the buoyancy of the pocket prevents it from being dragged downstream [7–9]. These large air pocket volumes can behave as air cushions and absorb the energy of transient pressure waves [10–17]. In the same manner, Qiu [18], Gahan [19] and He [20] demonstrated through numerical studies that large air pockets can result in a positive effect during a hydraulic transient event with in a pumping pipeline, replicating the behavior of air chambers.

Contrary to the effect of large air pockets in fluid transients, several researchers have demonstrated that the presence of small air pockets in pipelines can significantly increase the maximum pressures during hydraulic transients, enough to cause the failure of the pipe [21–29]. The magnitude of the damage will depend on the amount and location where the undissolved air is located, the configuration of the conduction, as well as the causes that generate the transient [30,31]. It is worth noting that Gahan [19] stated, after conducting an extensive and detailed review of the investigations related to

hydraulic transients with entrained air, that the criterion that establishes whether an air pocket is large or small will depend on its effect on transients.

This paper presents a case study of an existing pumping pipeline with five pumping plants each equipped with five centrifugal pumps and a backup pump connected in parallel. In order to study the pipeline behavior before put the system into operation, researchers of the Institute of Engineering of the University of Mexico developed several transient simulations for different scenarios and flow conditions. They found that the devices for surge protection of the investigated pipeline were insufficient to mitigate the adverse effects of transient events because the moment of inertia of all the pumps was erroneously considered by others during the design stage. Furthermore, the surge modeling results revealed that the most serious situation occurred when a simultaneous failure of the five pumps occurs at each pumping plant due to electric power cut, which could lead to water-column separation at some pipeline segments of the system. The latter compromised the pipeline safety and reliability. Therefore, the five pumps of each plant were never put into operation until the system for transient control was improved. The necessity to start providing water to the population led to attempt an unconventional and cost-effective form of protecting the line against low pressure transients. The solution was to operate two of the five units per plant, and allow air to enter through air/vacuum valves located along the pipeline. The air entrained formed pockets that remained stationary at high points of the line; these pockets behaved as air cushions that absorbed the energy of transient pressure waves. Computational simulations were conducted considering that two pumps are in operation at each plant and suddenly these fail simultaneously caused by power failure. Furthermore, the surge modelling results were compared with those registered during field pressure measurements.

## 2. The Pumping Pipeline System

Located in north Mexico, the pumping pipeline investigated has a total length of 90 km, was constructed of steel pipe with an inner diameter of 2.51 m (99 in) and its total head difference is 326 m. Five pumping plants (PP1 to PP5) transport water from the dam to the water treatment plant. Each pumping station is equipped with a cylindrical suction reservoir with a diameter of 50 m and 11 m high, and five centrifugal pumps and a backup pump connected in parallel to deliver a total flow rate of about 6.0 m$^3$/s 24 h a day, 365 days a year. In addition, each pump is fitted with an air/vacuum valve and a butterfly valve at its discharge. Figure 1 shows the pipeline profile.

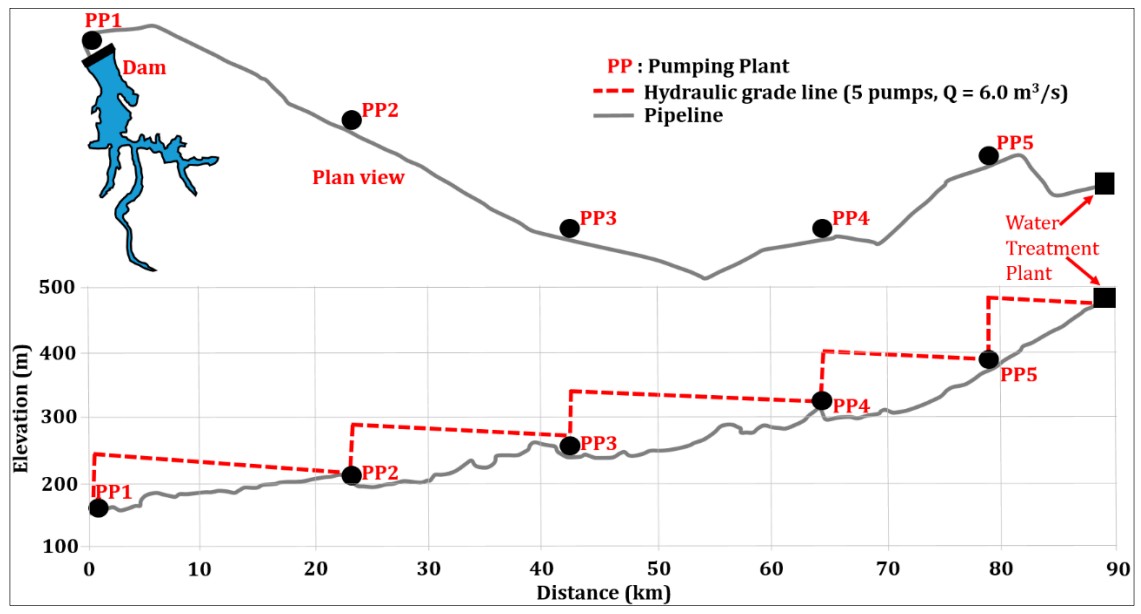

**Figure 1.** Profile of the pipeline.

The original system for control of hydraulic transients is comprised of eleven one-way surge tanks ranging from 17 m to 34 m in height and 4.5 m to 7.0 m in diameter, as well as three open surge tanks ranging from 11 m to 20 m high and 4.1 m to 6.0 m in diameter. Table 1 summarizes the main characteristics of the five pipeline segments.

**Table 1.** Characteristics of the five pipeline segments.

| Pumping Plant | Length (m) | Head (m) | One-Way Surge Tank | Open Surge Tank |
|---|---|---|---|---|
| PP1 | 23,200 | 80.12 | 3 | 1 |
| PP2 | 18,200 | 81.82 | 2 | 1 |
| PP3 | 23,000 | 79.32 | 2 | — |
| PP4 | 15,700 | 83.42 | 2 | 1 |
| PP5 | 9800 | 100.76 | 2 | — |

The Institute of Engineering of the University of Mexico (IE-UNAM) was commissioned by the pipeline owner with the main aim of supporting on the operations to start the pumping system. Likewise, it is important to highlight that the authors of this research were not involved in the design of the system under study. In order to analyze the pipeline behavior for multiple scenarios, researchers of the IE-UNAM performed a hydraulic transient analysis by using a numerical model based on the method of characteristics. From the computations obtained, it was found that the most serious situation occurred when a simultaneous failure of the five pumps occurs at each pumping plant due to electric power cut. Likewise, it is important to highlight that, as a result of the numerical analyses, it was found that the system for control of fluid transients was insufficient because the pipeline designer incorrectly considered the moment of inertia of all the pumps, which significantly compromised the pipeline safety and reliability. According to the original design, the moment of inertia of the pumps used for dimensioning the surge control devices was up to 4.5 times greater than that specified by the manufacturer, as shown in Table 2. Therefore, additional devices for transient control will be required to minimize the maximum head and to maximize the minimum head in the system to within acceptable limits.

**Table 2.** Moment of inertia of the pumps.

| Pumping Plant | Manufacturer Data | | | Data Used in the Original Design |
|---|---|---|---|---|
| | Pump (Kg-m$^2$) | Motor (Kg-m$^2$) | Total (Kg-m$^2$) | Total (Kg-m$^2$) |
| PP1 | 13.19 | 101.25 | 114.44 | 460 |
| PP2 | 15.00 | 98.75 | 113.75 | 510 |
| PP3 | 15.00 | 98.75 | 113.75 | 510 |
| PP4 | 15.00 | 98.75 | 113.75 | 510 |
| PP5 | 22.00 | 103.75 | 125.75 | 510 |

The pump inertia is used to evaluate the necessary starting torque of the motor during a normal start-up and for calculating its coast-down speed once the power is turned off to the motor. The latter is used for the hydraulic transient analysis to determine the severity of the transient pressures when the pump is switched off [6]. Large pumps have more inertia than small pumps, since they have more rotating mass. A pump with higher inertia can assist with controlling transients in order for the rotating parts of the pump and motor continue moving water through the pump for a longer time as they slowly decelerate following a power failure [32]. However, when pumping equipment has a limited inertia of rotating parts, the pump speed will decrease very rapidly, generating negative pressure waves, which could lead to water-column separation [4]. Therefore, an adequate surge protection system should always be considered during the design stage of pumping pipelines.

## 3. Numerical Model

The analysis of the effect of air pocket volumes introduced into the investigated pumping pipeline via the air valves was developed based on the method of characteristics (MOC) [4,33]. Numerical models based on the MOC are known to give accurate results and have demonstrated to be effective [34–36]. They have been successfully applied in the study of pumping pipelines involving air pockets [23,27,37].

Combination air valves were installed at high points throughout the pipeline under study with the aim of assisting in pipeline filling operation and protecting the system from low pressures that may be caused by water-column separation, pipeline draining, pump failure, or a pipe rupture. Although combination air valves expel the air from pipelines, during the hydraulic transient analysis, it is considered that air entrained through the valves is entrapped and not allowed to release when the pressure increases above the atmospheric pressure, since the air release through the small orifice is very slowly. Likewise, it is taken into account that the entrapped air form pockets that remain at the valve location and are not removed by the flowing water.

Furthermore, the general gas equation was used for analyzing the behavior of the air pockets in the pipeline. This equation may be written as

$$p^*V = mRT, \tag{1}$$

where $V$ and $m$ are the volume and mass of the air pocket, $p^*$ and $T$ are the absolute pressure and temperature of the air pocket, and $R$ is the universal gas constant.

Since in this investigation the MOC is used to analyze the effect of air pockets on hydraulic transients, the positive and negative characteristic equations at the end of each time interval for sections $(i, N + 1)$ and $(i + 1, 1)$ are defined as follows:

$$QU_{i,N+1} = C_p - B_{a_i}HU_{i,N+1}, \tag{2}$$

$$QU_{i+1,1} = C_n + B_{a_{i+1}}HU_{i+1,1}, \tag{3}$$

where

$$C_p = Q_{i,N+1} + B_{a_i}H_{i,N+1} - R_iQ_{i,N+1}|Q_{i,N+1}|, \tag{4}$$

$$C_n = Q_{i+1,1} - B_{a_{i+1}}H_{i+1,1} - R_{i+1}Q_{i+1,1}|Q_{i+1,1}|, \tag{5}$$

$$B_{a_i} = \frac{gA_i}{a_i}, \tag{6}$$

$$B_{a_{i+1}} = \frac{gA_{i+1}}{a_{i+1}}, \tag{7}$$

$$R_i = \frac{f_i\Delta t_i}{2D_iA_i}, \tag{8}$$

$$R_{i+1} = \frac{f_{i+1}\Delta t_{i+1}}{2D_{i+1}A_{i+1}}, \tag{9}$$

in which $Q_{i,N+1}$ and $QU_{i,N+1}$ are the water flow discharges at the upstream end of the pocket at the beginning and end of the time step, respectively; $Q_{i+1,1}$ and $QU_{i+1,1}$ are the water flow discharges at the downstream end of the air pocket at the beginning and end of the time step, respectively. $A$ is the total cross section area of the pipe, $g$ is the gravitational acceleration, $a$ is the transient wave speed, $f$ is the Darcy–Weisbach friction factor, $D$ is the pipe diameter, and $\Delta t$ is the time step.

The finite difference scheme remains stable due to the fact that the Courant–Friedrich–Lewy condition is satisfied during all the surge modelling:

$$\Delta x \geq a\Delta t, \tag{10}$$

where $\Delta x$ is the pipe reach length.

Moreover, if the head losses at the air valve and junction are neglected, then

$$H_{U_{i,n+1}} = H_{U_{i+1,1}}. \tag{11}$$

The subscript $U$ is used to denote the variables that are unknown at the end of the time step, whereas the variables without the subscript $U$ are known quantities from the previous time step. Likewise, for the junctions $(i, N + 1)$ and $(i + 1, 1)$, the first subscript defines the conduit number, and the second designates the section number. Figure 2 shows the notation for the air pocket.

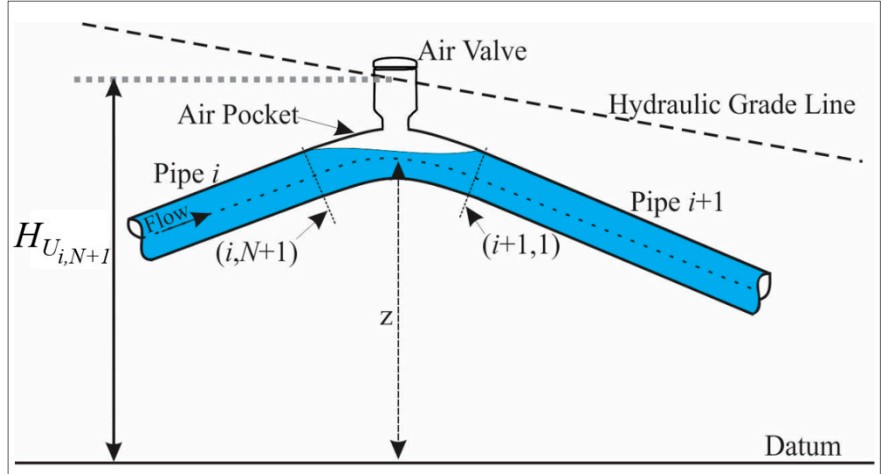

**Figure 2.** Notation for the air pocket.

During transient simulations, the airflow into the pipeline is isentropic and the mass of the air pocket in the pipeline at the beginning of the time step is designated by $m_i$, then the mass of air at the end of time step $m_{U_{i,N+1}}$ may be written as (Chaudhry [4]):

$$m_{U_{i,N+1}} = m_i + \frac{dm_i}{dt}\Delta t, \tag{12}$$

where $\frac{dm_i}{dt}$ is the time rate of mass inflow of air through the valve into the pipeline.

In the same way, the compression and expansion of air pockets can be properly simulated by considering the isothermal process. This phenomenon can be represented mathematically as:

$$p^* V_{U_{i,N+1}} = m_{U_{i,N+1}} RT. \tag{13}$$

The continuity equation for the air pocket can be presented as:

$$V_{U_{i,N+1}} = V_i + \frac{\Delta t}{2}\left[\left(Q_{U_{i+1,1}} + Q_{i+1,1}\right) - \left(Q_{U_{i,N+1}} + Q_{i,N+1}\right)\right], \tag{14}$$

where $V_i$ and $V_{U_{i,N+1}}$ are the air pocket volume at the beginning and end of the time step, respectively.

By substituting Equations (2) through (11) into Equation (14),

$$V_{U_{i,N+1}} = C_{air} + \frac{\Delta t}{2}\left[\left(B_{a_i} + B_{a_{i+1}}\right)H_{U_{i,N+1}}\right], \tag{15}$$

is obtained where

$$C_{air} = V_i + \frac{\Delta t}{2}\left(Q_{i+1,1} - Q_{i,N+1} + C_n - C_p\right). \tag{16}$$

The numerical model calculates the total head $H_{U_{i,N+1}}$, which is related with the absolute pressure $p^*$ through the equation

$$p* = \gamma\left(H_{U_{i,N+1}} - z_{i,N+1} + H_{Atm}\right), \tag{17}$$

where $z_{i,N+1}$ is the height of the air valve above the datum, $\gamma$ is the specific weight of water and $H_{Atm}$ is the atmospheric pressure head.

Substituting $H_{U_{i,N+1}}$ from Equation (17) into Equation (15), eliminating $V_{U_{i,N+1}}$ from the resulting equation and Equation (13), gives

$$m_{U_{i,N+1}}RT = p^*\left\{C_{air} + \frac{\Delta t}{2}\left[\left(B_{a_i} + B_{a_{i+1}}\right)\left(\frac{p^*}{\gamma} + z_{U_{i,N+1}} - H_{Atm}\right)\right]\right\}. \tag{18}$$

Elimination of $m_{U_{i,N+1}}$ from Equations (12) and (18) yields

$$\left(m_i + \frac{dm_i}{dt}\right)\Delta t RT = p^*\left\{C_{air} + \frac{\Delta t}{2}\left[\left(B_{a_i} + B_{a_{i+1}}\right)\left(\frac{p^*}{\gamma} + z_{U_{i,N+1}} - H_{Atm}\right)\right]\right\}. \tag{19}$$

In the above equation, $p^*$ and $\frac{dm_i}{dt}$ are the two unknowns. Whether the absolute pressure, $p^*$, within the pipeline is lower than $0.53p_a$ ($p_a$ = atmospheric pressure), the air velocity through the valve is sonic, whereas if $p^*$ is larger than $0.53p_a$ but lower than $p_a$, the airflow through the valve is at subsonic velocity. The equations for $\frac{dm_i}{dt}$ are (Streeter [38]):

Subsonic air velocity through the valve ($p_a > p^* > 0.53p_a$)

$$\frac{dm_i}{dt} = C_d A_v \sqrt{7 p_a \rho_a \left(\frac{p^*}{p_a}\right)^{1.43}\left[1 - \left(\frac{p^*}{p_a}\right)^{0.286}\right]}. \tag{20}$$

Sonic air velocity through the valve ($p^* \leq 0.53p_a$)

$$\frac{dm_i}{dt} = 0.686 C_d A_v \frac{p_a}{\sqrt{RT_a}}, \tag{21}$$

where $C_d$ is the valve discharge coefficient, $A_v$ is the area of the valve opening at its throat; $\rho_a$ is the mass density of air at atmospheric pressure and absolute temperature $T_a$ outside the pipeline.

In the same way, once Equations (20) or (21) is substituted into Equation (19) results a nonlinear equation in $p^*$, which may be resolved by an iterative method, for instance, the Newton–Raphson method. Likewise, the values of the unknown variables $H_{U_{i,N+1}}$, $V_{U_{i,N+1}}$, $m_{U_{i,N+1}}$, $Q_{U_{i,N+1}}$, and $Q_{U_{i+1,1}}$ may be evaluated from Equations (2) through (18).

In addition, it is important to highlight that the numerical model has the capability of simulating transient control devices (air chambers, surge tanks, one-way surge tanks, etc.) for typical pump operations, such as pump start-up, pump shut-down, unexpected loss of electric power on the pumps, etc.

## 4. Hydraulic Transient Simulations

### 4.1. Transient Flow Analysis for Five Pumps in Operation at Each Station

Several transient simulations for normal maneuvers and unplanned situations were conducted by using the numerical model with the purpose of finding the worst-case scenario, when the five pumping plants of the pipeline investigated operate with five pumps. Since various researchers state that the most serious situation in a pumping station occurred when all the pumping devices fail simultaneously caused by power failure [4,6,11], this situation was first simulated by considering the incorrect moment of inertia of the pumps used in the original design.

In the same way, the simultaneous failure of the five units at the five pumping stations was computed taking into account the moment of inertia provided by the pump manufacturer, which is

the correct moment of inertia (see Table 2). From the calculations obtained, it was noticed that the devices for transient control do not provide adequate surge protection against transient pressures, due to the moment of inertia of all the pumps being incorrectly considered during the design stage, which significantly compromised the pipeline safety and reliability. According to the original design, the moment of inertia of the pumps used for dimensioning the system for control of hydraulic transients was up to 4.5 times greater than that specified by the manufacturer. Therefore, additional devices for transient control will be necessary to avoid catastrophic damage to the pipeline.

In order to protect adequately the pumping system, a detailed hydraulic transient analysis was performed for the design and testing of additional surge suppression devices, assuming the sudden power failure to the pumps, when the pipeline is working at its maximum capacity of 6 m³/s. The devices contemplated were air chambers, one-way surge tanks and open surge tanks. Furthermore, transient simulations were conducted by considering the various surge protection options, as well as the correct moment of inertia of the pumps. The surge modeling results indicated that the most efficient measure to minimize the upsurge pressures and to maximize the downsurge pressures in the system to within acceptable limits is to install three air chambers at the discharge header of each pumping station to complement the existing system for transient control.

During the hydraulic transient simulations conducted over all the investigation, the celerity remains constant $a = 950$ m/s. Likewise, the primary characteristics of the pumps and the air chambers are summarized in Table 3.

**Table 3.** Characteristics of the pumps and the air chambers.

| Pumping Plant | Rated Discharge (m³/s) | Rated Head (m) | Rated Speed (rpm) | Number of Air Chambers | Diameter of Air Chambers (m) | Height of Air Chambers (m) | Diameter of Connector Pipe (m) | Maximum Pressure (mH$_2$O) |
|---|---|---|---|---|---|---|---|---|
| PP1 | 1.4 | 80.12 | 1180 | 3 | 3.37 | 9.24 | 1.066 (42 in) | 95 |
| PP2 | 1.2 | 81.82 | 1180 | 3 | 2.34 | 5.27 | 0.914 (36 in) | 110 |
| PP3 | 1.2 | 79.32 | 1180 | 3 | 2.34 | 4.57 | 0.914 (36 in) | 110 |
| PP4 | 1.2 | 83.42 | 1180 | 3 | 2.34 | 8.55 | 0.914 (36 in) | 175 |
| PP5 | 1.2 | 100.76 | 1180 | 3 | 3.37 | 5.75 | 1.066 (42 in) | 150 |

### 4.2. Transient Flow Analysis for Two Pumps in Operation at Each Station

Since the manufacturing process and installation of the three air chambers at each of the pumping plants would take some time and, due to the necessity to start supplying water to the population as soon as possible, prompt the authors to find a cost-effective solution that could be easily and rapidly implemented, but also offering an adequate protection to the pipeline against transient pressures. With this purpose, preliminary numerical simulation was developed. The system was analyzed for the sudden power failure to the pump stations with one, two, three and four pumps in operation with its correct moment of inertia. In all instances, the results indicated that the minimum head enveloped intersects some sections of the pipeline profile, producing negative pressures. Therefore, it can be stated that the original system for transient control, as well as the existing air valves, are unable of mitigating transient pressures caused by power failure, regardless of the number of pumps in operation.

Based on the above, the authors suggested an unconventional form of protecting the pipeline against low pressures to put the system into operation with less than five pumping units per plant. The temporary solution proposed was to allow large quantities of air to enter the pipeline through air valves, when the hydraulic grade line falls below its elevation. When this occurs, large air pockets remain stationary at high points of the line because the buoyancy of the pockets prevent them from being dragged downstream by the drag force of water flow [7–9]. Likewise, it is well known that the effect of large pockets on transient pressures can be beneficial [10–17]. Moreover, various researchers have demonstrated that large air pockets can act as effective accumulators suppressing the energy of pressure waves, replicating the behavior of air chambers [18–20].

Subsequently, numerous hydraulic transient simulations caused by power failure to pumps were conducted to select and size more appropriate air valves in order to enter the necessary volume of air to the pipeline to ensure that large air pockets remain stationary in adequate locations. Results show the best solution is to operate two of the five units per plant with the original control devices and install additional air valves at strategic locations to prevent the occurrence of negative pressures. Likewise, it is important to highlight that it is not a common practice to admit large quantities of air into pipelines of large dimensions to avoid negative pressures [4,11].

The existing and additional combination air valves located along the investigated pipeline are advanced devices that combine the air release and vacuum break valves in a single body. These valves have small precision orifices to exhaust air, whereas the pipeline is in service, and the large orifices diameters equal the nominal size of the valves to diminish the resistance to the air intake and decreasing significantly the potential negative pressure within the pipeline during a draining operation, pipeline rupture, or pump failure. Likewise, the valve design guarantees the effective elimination of all air and the cylindrical solid floats made of High-density polyethylene (HDPE) reduce the risk for dynamic closure while eradicating the possibilities of water hammer on closure of the large orifice.

It is worth mentioning that, if not appropriately sized, air valves can worsen the transient response of the system. For instance, Lee [39] stated that the efficacy of the air valves for transient protection hinges not only on the pipeline system configuration, the physical properties of the pipeline and the fluids, but significantly also on the characteristics of the air valves, as well as on the distribution of air pocket volumes in the system concerned. The author also demonstrated that air valves with high inflow characteristics located at high points of a pipeline with entrapped air may decrease the magnitude of the extreme negative pressures. On the other hand, air valves with higher outflow characteristics tend to result in higher positive pressures. Therefore, the appropriate sizing of air valves is important for effective, efficient, and safe air control. Conversely, the incorrect sizing of air valves may cause large pressure peaks immediately following the rapid expulsion of an air pocket [40–43].

Finally, it is important to emphasize that, once air is admitted into the pipeline, caution must be exercised for venting air during refilling the pipeline. The air should be released slowly from the pipeline because the entrapped air may result in very high pressures [4].

Table 4 summarized the location and dimensions of the existing and additional air valves installed throughout the five pipeline segments, as well as the air pocket volumes obtained with Equation (15).

### 4.3. Field Measurements for the Simultaneous Failure of Two Pumps at Each Station

After the additional air valves were installed along the pumping pipeline, field pressure measurements were conducted at different locations throughout the system. At the start of each field test, two pumps were operated at each plant to supply a water discharge of approximately 2.4 m$^3$/s. Subsequently, the pumping plant operators were instructed to switch off the operating pumps simultaneously. This maneuver was equivalent to a sudden loss of power to the pumping station. The transient pressures were registered at the downstream end of the discharge header of the five pumping stations and at various locations of the air valves.

High-frequency pressure transducers were used during the pressure measurements in the pipeline. These transducers are capable of detecting pressures between $-10.4$ mH$_2$O ($-14.7$ psi) vacuum and 351 mH$_2$O (500 psi) with an accuracy of ±0.5%. The data acquisition frequency was set at 10 Hz. Data loggers registered the signals from the pressure transducers and achieved a digital conversion for direct storage in the hard disk of a laptop computer for later analysis.

During the field measurements, the initial steady-state water discharge in the pumping pipeline was measured by means of ultrasonic flowmeters with an accuracy of ± 0.5%. These devices were installed on the discharge headers of the pumping stations.

The transient pressure data from the field tests were compared to the results achieved in the hydraulic transient analysis, with the main purpose of verifying the reliability of the computational

program used during the present investigation. The comparison between the results from the field measurements and those obtained with the numerical model are reported within the next section.

**Table 4.** Location and dimensions of air valves and air pocket volumes, (*) denotes the new air valves installed in the pipeline.

| Chainage (m) | Elevation (m) | Diameter of Air Valve (in) | Number of Air Valves | Air Pocket Volumes (m$^3$) |
|---|---|---|---|---|
| First pipeline segment (from PP1 to PP2) | | | | |
| 2071 | 175.10 | 6 | 1 | 0.870 |
| 4764 (*) | 175.00 | 6 | 1 | 0.240 |
| 4917 (*) | 180.00 | 6 | 1 | 0.620 |
| 9780 | 187.60 | 6 | 1 | 0.336 |
| 10,608 | 188.96 | 6 | 1 | 0.330 |
| 11,790 (*) | 191.49 | 6 | 1 | 0.626 |
| 12,626 | 191.16 | 4 | 2 | 0.183 |
| 13,045 | 191.08 | 6 | 1 | 0.267 |
| 13,400 (*) | 193.75 | 6 | 1 | 0.308 |
| 13,600 (*) | 196.75 | 4 | 1 | 1.044 |
| 18,706 | 206.87 | 6 | 2 | 0.591 |
| 19,720 | 208.02 | 4 | 2 | 0.311 |
| 20,760 | 210.03 | 4 | 1 | 0.474 |
| 21,920 | 216.68 | 6 | 2 | 1.335 |
| Second pipeline segment (from PP2 to PP3) | | | | |
| 30,684 | 222.57 | 4 | 1 | 0.756 |
| 33,300 (*) | 237.87 | 4 | 2 | 0.624 |
| 33,564 | 243.62 | 6 | 1 | 0.204 |
| 38,650 (*) | 261.10 | 6 | 1 | 0.312 |
| 38,800 | 269.03 | 6 | 1 | 1.193 |
| Third pipeline segment (from PP3 to PP4) | | | | |
| 53,400 (*) | 263.75 | 6 | 1 | 0.414 |
| 53,770 | 267.19 | 4 | 2 | 0.476 |
| 54,215 (*) | 273.86 | 6 | 1 | 0.528 |
| 54,560 | 283.73 | 6 | 2 | 0.138 |
| 55,560 | 292.38 | 6 | 1 | 0.189 |
| 63,317 (*) | 305.77 | 6 | 1 | 0.252 |
| 63,417 (*) | 310.06 | 6 | 2 | 0.830 |
| 63,627 (*) | 315.52 | 6 | 1 | 0.642 |
| 63,737 | 318.80 | 6 | 1 | 10179 |
| Fourth pipeline segment (from PP4 to PP5) | | | | |
| 69,116 | 317.52 | 4 | 2 | 0.252 |
| 72,016 | 319.05 | 6 | 1 | 0.416 |
| 72,696 | 321.75 | 6 | 1 | 0.522 |
| 73,086 (*) | 324.91 | 6 | 1 | 0.464 |
| 73,476 | 329.71 | 4 | 2 | 0.758 |
| 73,826 (*) | 335.79 | 6 | 1 | 0.512 |
| 74,036 | 342.80 | 4 | 1 | 0.621 |
| 74,236 | 346.98 | 6 | 2 | 0.596 |
| 76,666 | 373.80 | 6 | 2 | 0.471 |
| 77,736 (*) | 375.66 | 6 | 1 | 0.468 |
| 77,926 | 379.38 | 6 | 2 | 0.904 |
| Fifth pipeline segment (from PP5 to Water treatment plant) | | | | |
| 80,579 (*) | 399.06 | 6 | 1 | 0.526 |
| 80,967 | 405.04 | 4 | 2 | 0.782 |
| 81,339 (*) | 409.85 | 6 | 2 | 0.872 |
| 81,727 | 413.96 | 6 | 1 | 0.774 |
| 82,107 (*) | 417.54 | 6 | 1 | 0.766 |
| 82,487 | 420.96 | 6 | 1 | 0.654 |
| 84,047 (*) | 438.76 | 6 | 1 | 0.687 |
| 84,447 | 443.54 | 6 | 1 | 0.574 |
| 84,774 | 447.49 | 4 | 2 | 0.792 |
| 85,024 | 449.53 | 6 | 2 | 0.870 |
| 86,800 (*) | 463.17 | 6 | 1 | 0.514 |
| 86,967 | 467.66 | 4 | 2 | 0.668 |
| 87,314 | 475.66 | 6 | 2 | 1.307 |
| 87,617 | 474.81 | 6 | 2 | 1.597 |

## 5. Results and Discussion

For brevity, only the hydraulic transient results associated with the simultaneous failure of five and two pumps in the pumping stations due to electric power cut are presented.

*5.1. Transient Pressures Caused by the Simultaneous Failure of Five Pumps at Each Station*

Figures 3–7 show the maximum and minimum head envelopes obtained from the numerical model by considering the incorrect moment of inertia of the pumps used during the design stage, the correct moment of inertia provided by the pump manufacturer, as well as the correct moment of inertia and the three additional air chambers installed at the discharge header of each pumping station.

The results achieved by taking into account the incorrect moment of inertia of the pumps used in the original design show a suitable design of the pumping pipeline, as can be observed in Figures 3–7. In contrast, if the correct moment of inertia is considered, results revealed the occurrence of negative transient pressures at the five pipeline segments, since the system for control of hydraulic transients of the pipeline was insufficient to adequately suppress downsurge pressures, due to the moment of inertia of all pumps being erroneously considered during the design stage. According to the original design, the moment of inertia of the pumps used for dimensioning the surge suppression devices was up to 4.5 times greater than that specified by the pump manufacturer.

The maximum and minimum head envelopes obtained regarding the moment of inertia provided by the pump manufacturer show a noticeable increase of the transient pressures along the system, in comparison with the surge modeling results achieved with the erroneous moment of inertia. It is clearly observed that the problem of negative pressures is generated by the great difference between the correct moment of inertia and that used in the design, since the moment of inertia is one of the parameters with the greatest impact on hydraulic transients caused by power failure to pumps. Consequently, the calculated minimum pressures are much lower than those specified in the design.

When the correct moment of inertia is considered, it is observed that the pipeline will experience negative pressure waves which travel along the system, causing the pressure in the five pipeline segments to fall rapidly, creating sub-atmospheric pressure over sections of the pipeline. Likewise, the minimum head envelopes reveal that the transient pressures at the first, third and fifth pipeline segments may be reduced to the vapor pressure of the liquid, which could lead to cavitation and water column separation following a loss of power to the pumps. This would take place between the chainages 13 + 600 to 13 + 840, 73 + 836 to 74 + 336, 80 + 918 to 82 + 838, 84 + 918 to 85 + 367, and 87 + 339 to 88 + 279, respectively. Furthermore, after re-pressurization of the pipeline by a flow reversal or a reflected transient pressure wave, any vapor cavities that have formed would collapse and could create high-localized pressure spikes that may damage the pipeline. Therefore, it was of paramount importance to improve the existing system for transient control by adding new devices to uplift the minimum head envelopes in the system to within acceptable limits.

The results show that the proposed control devices provide a safe and stable hydraulic condition and protect the system against severe transient pressures. It can be observed in Figures 3–7 that, after the simultaneous failure of the five pumps at each station, the maximum and minimum heads along the pumping pipeline were considerably reduced by installing the three air chambers at the discharge header of each pumping station. In this case, the cushioning effect produced by the air chambers absorbed the transient pressures waves considerably. Therefore, it can be stated that these devices have a beneficial effect over the entire the pumping pipeline system.

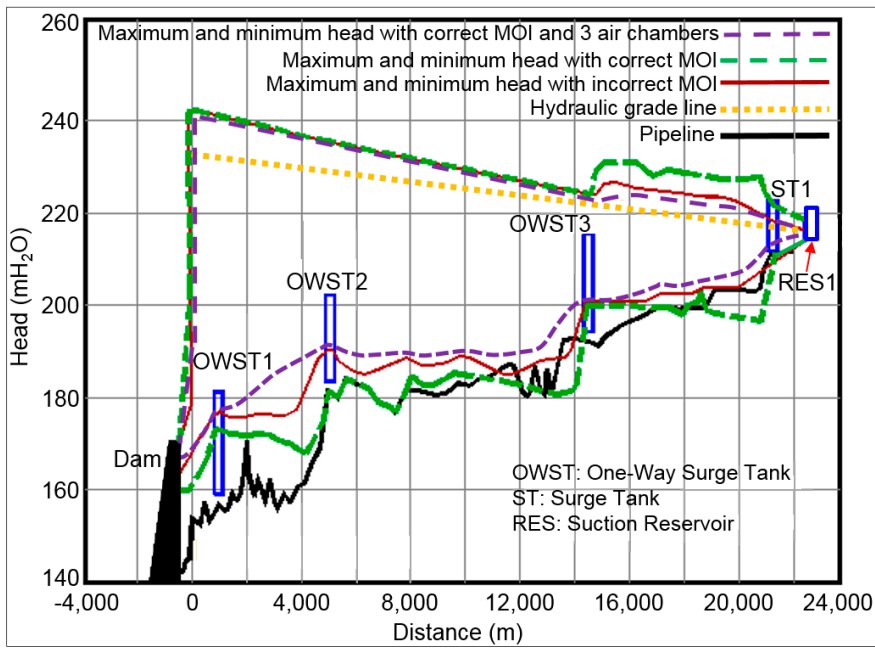

**Figure 3.** Maximum and minimum head envelopes for the first pipeline segment, considering the incorrect moment of inertia (original design), the correct moment of inertia (MOI), and the correct moment of inertia and the three additional air chambers, following the simultaneous power failure of the five pumps at Pumping Plant 1.

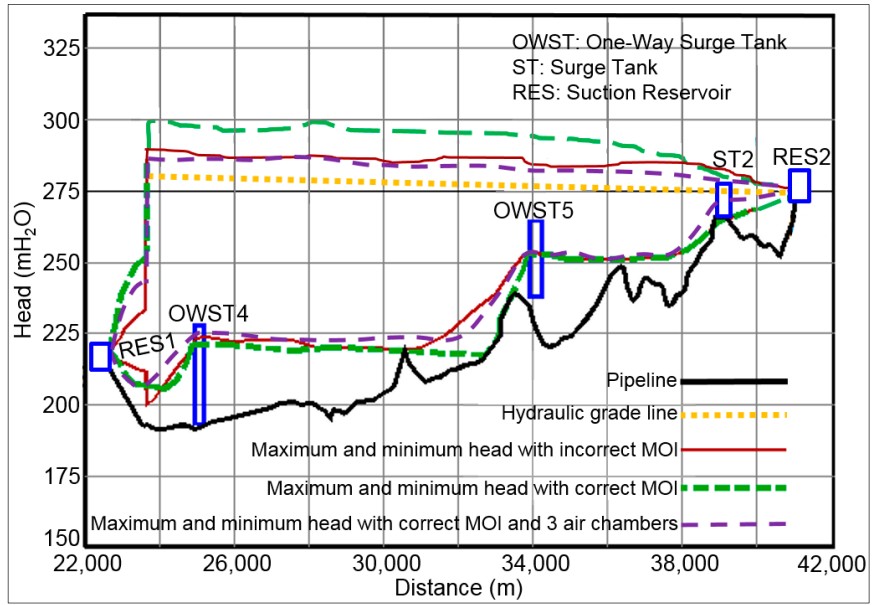

**Figure 4.** Maximum and minimum head envelopes for the second pipeline segment, considering the incorrect moment of inertia (original design), the correct moment of inertia (MOI), and the correct moment of inertia and the three additional air chambers, following the simultaneous power failure of the five pumps at Pumping Plant 2.

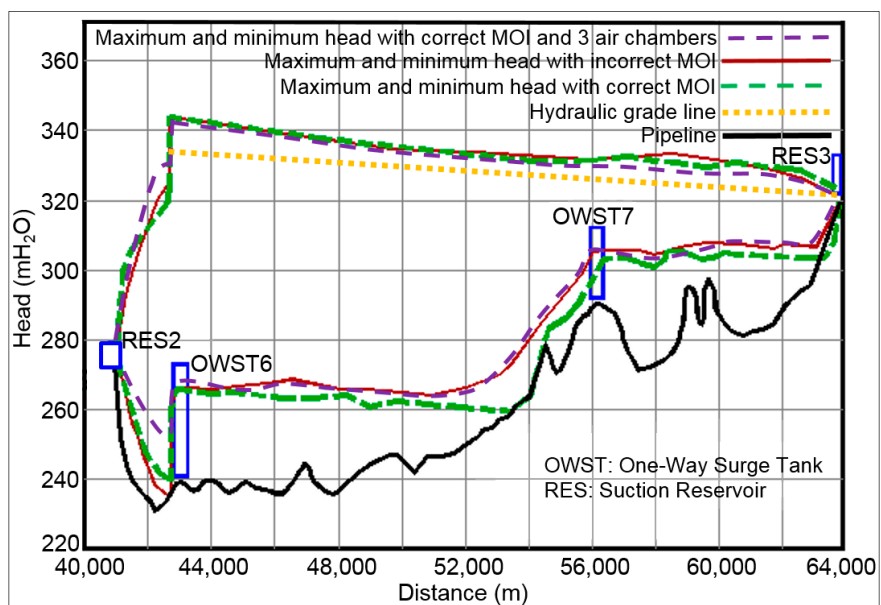

**Figure 5.** Maximum and minimum head envelopes for the third pipeline segment, considering the incorrect moment of inertia (original design), the correct moment of inertia (MOI), and the correct moment of inertia and the three additional air chambers, following the simultaneous power failure of the five pumps at Pumping Plant 3.

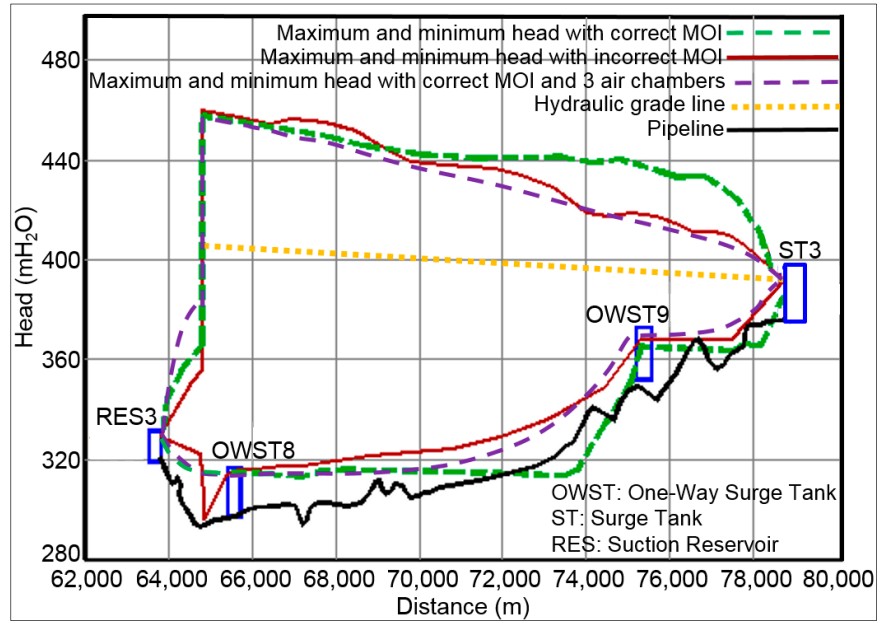

**Figure 6.** Maximum and minimum head envelopes for the fourth pipeline segment, considering the incorrect moment of inertia (original design), the correct moment of inertia (MOI), and the correct moment of inertia and the three additional air chambers, following the simultaneous power failure of the five pumps at Pumping Plant 4.

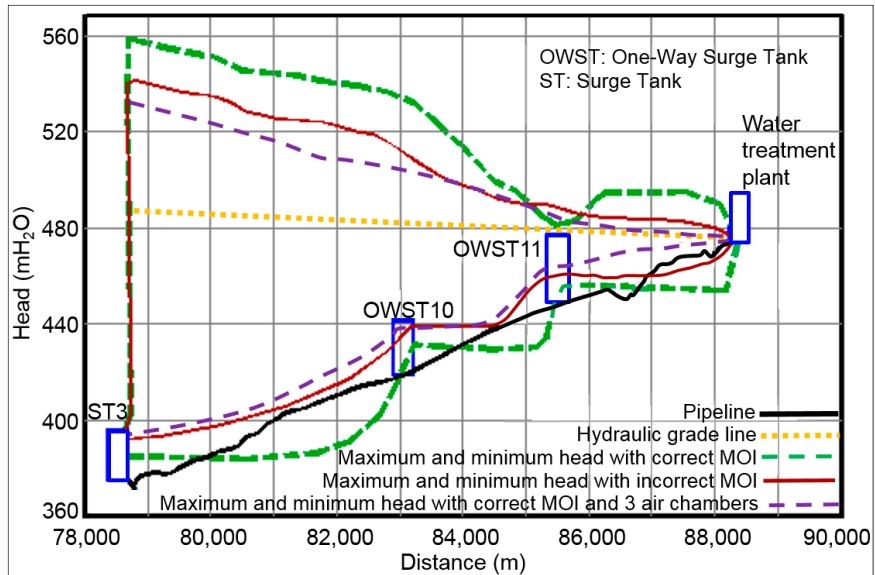

**Figure 7.** Maximum and minimum head envelopes for the fifth pipeline segment, considering the incorrect moment of inertia (original design), the correct moment of inertia (MOI), and the correct moment of inertia and the three additional air chambers, following the simultaneous power failure of the five pumps at Pumping Plant 5.

### 5.2. Transient Pressures Caused by the Simultaneous Failure of Two Pumps at Each Station

The best option to begin supplying water to the population while the air chambers are manufactured and installed at the discharge header of each pumping station is to operate two of the five pumps per station and allow air to enter through the existing and additional combination air valves installed at strategic locations to avoid surge damage to the pipeline, when the simultaneous failure of two pumps occurs at each pumping station. The air entrained formed large pockets that remained stationary at the valves locations; these pockets behaved as air cushions that absorbed the energy of transient pressure waves.

It is important to highlight that the original system for surge control and the existing combination air valves are insufficient for protecting the pipeline against the simultaneous failure of two pumps per station. It can be observed from Figures 8–12 that the pipeline will experience negative pressures at some high points or summits of the system based on the minimum head envelopes. Moreover, Figure 12 shows that most of the last pipeline segment would experience downsurge pressures, following the sudden power failure of the two pumps at the pumping plant.

On the other hand, Figures 8–12 also show the hydraulic transient results obtained by considering the additional and the existing combination air valves, as well as the original system for surge control. In this case, the maximum and minimum pressure transients along the pipeline are lower than those obtained without additional air valves. The large air pockets produced a cushioning effect, absorbing the transient pressure wave uplifting the minimum head and reducing the maximum head. It is worth mentioning that no negative pressures occurred in either pipeline segment. Therefore, it can be stated that the volumes of air entrained through the new and existing air valves were effective at preventing negative pressures. This temporary solution allowed for supplying to the population a water discharge of 2.4 $m^3$/s. Likewise, it is pertinent to mention that, to the authors' knowledge, this study presents one of the first applications of large air pockets to avoid negative pressures in a pipeline of large dimensions.

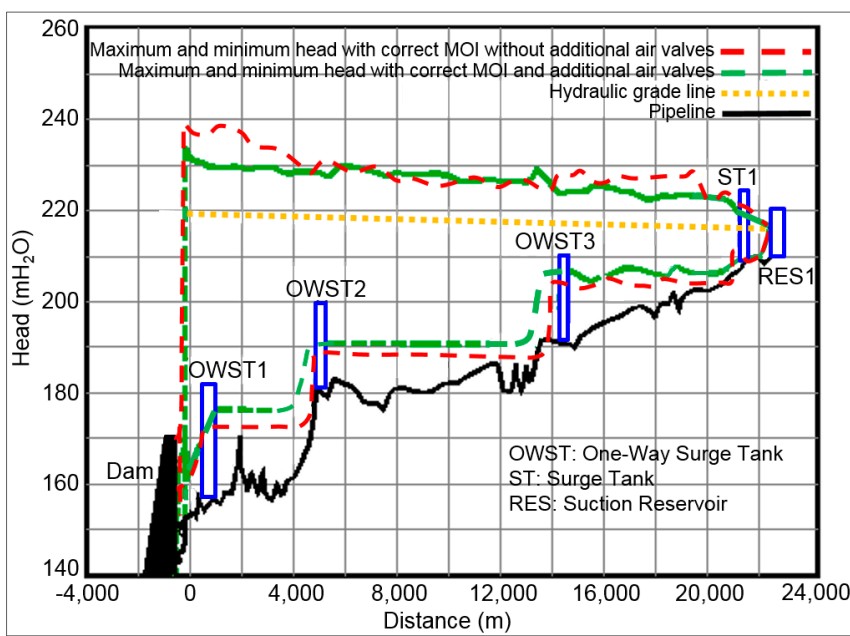

**Figure 8.** Maximum and minimum head envelopes for the first pipeline segment with and without additional air valves and the correct moment of inertia (MOI), following the simultaneous power failure of the two pumps at Pumping Plant 1.

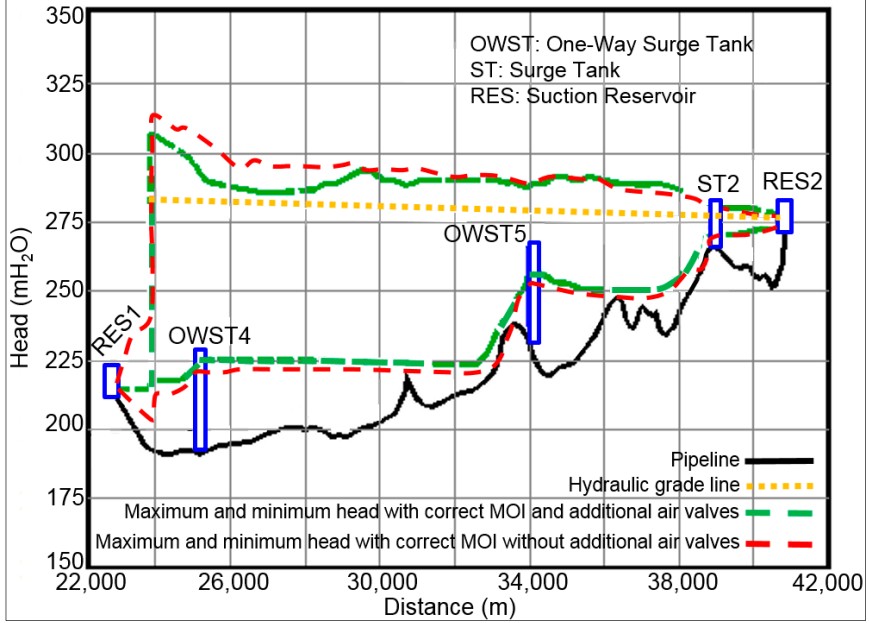

**Figure 9.** Maximum and minimum head envelopes for the second pipeline segment with and without additional air valves and the correct moment of inertia (MOI), following the simultaneous power failure of the two pumps at Pumping Plant 2.

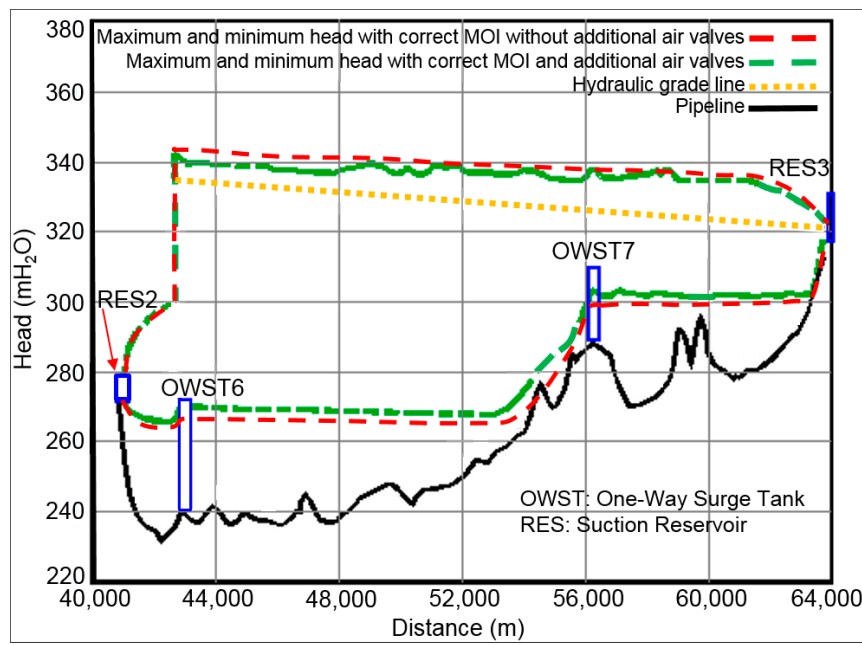

**Figure 10.** Maximum and minimum head envelopes for the third pipeline segment with and without additional air valves and the correct moment of inertia (MOI), following the simultaneous power failure of the two pumps at Pumping Plant 3.

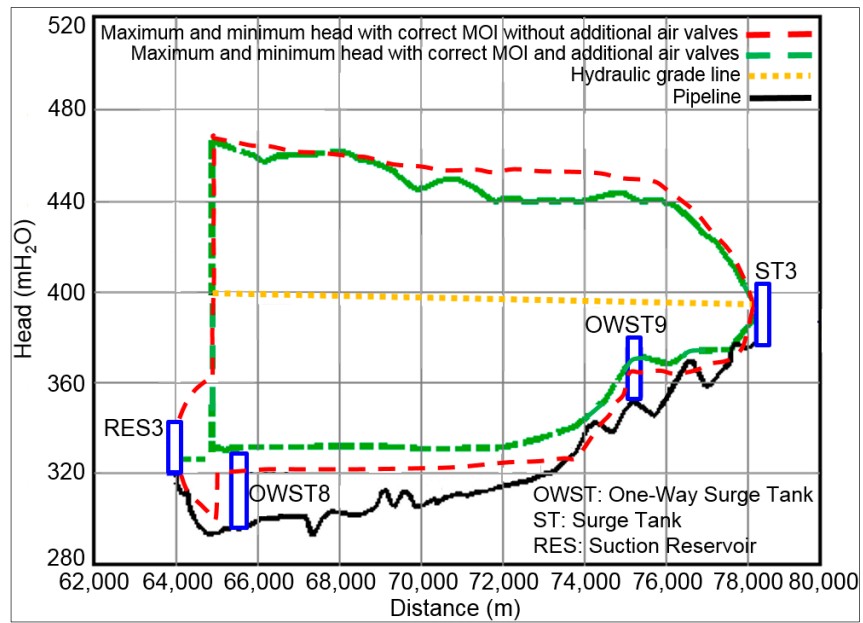

**Figure 11.** Maximum and minimum head envelopes for the fourth pipeline segment with and without additional air valves and the correct moment of inertia (MOI), following the simultaneous power failure of the two pumps at Pumping Plant 4.

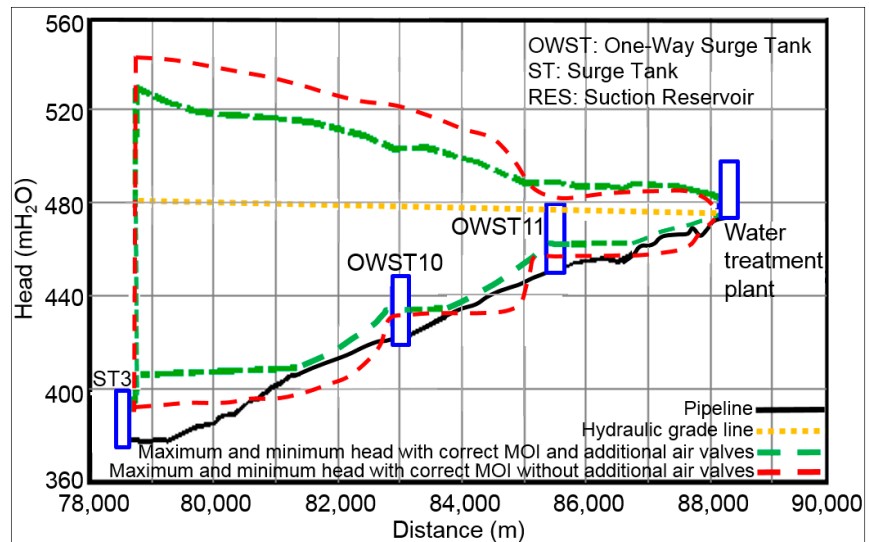

**Figure 12.** Maximum and minimum head envelopes for the fifth pipeline segment with and without additional air valves and the correct moment of inertia (MOI), following the simultaneous power failure of the two pumps at Pumping Plant 5.

*5.3. Comparison of Field Measurements with Numerical Results after Simultaneous Failure of Two Pumps at Each Station*

In order to compare the computed results obtained during the hydraulic transient simulations of the pumping pipeline system, field tests were performed after the additional combination air valves were installed. The field measurements consisted of recording transient pressures following the simultaneous failure of two pumps at each pumping station. The measurements were conducted at the downstream end of the discharge header of the five pumping stations and at various locations of the air valves.

Figures 13–17 show the pressure transient comparison of the simulated and field pressures. It can be observed that the pressure amplitudes calculated are slightly higher than the pressures recorded. Figures 13a, 14a, 15a, 16a and 17a show that the maximum pressure transients occur at the discharge header. Moreover, it can be seen from the modeled and the field pressures that, after simultaneous failure of two pumps per plant, the pressure falls, and, depending on the location of the air valves, the pressure flow drops approximately 6 to 17 seconds later, allowing air to enter through the air valves, as can be observed in Figures 13b, 14b, 15b, 16b and 17b. The air pockets produced a cushioning effect, absorbing the transient pressure wave and uplifting the minimum head. Furthermore, following the loss of power to the pumps at each station, the pressure falls, but, after a few seconds, the pressure starts increasing. It must be highlighted that, at the discharge headers and the locations of the air valves, the vapor pressure, which is the cavitation head of water at 20 °C ($-10.1$ mH$_2$O), was never reached. The lowest simulated and measure pressure was zero (0 mH$_2$O). Moreover, since pressure transients showed the same pattern, only part of the results are presented.

Based on the aforementioned, it can be stated that the model results are generally in good agreement with the field data. The slight differences in the minimum and maximum values are likely due to uncertainties of the input parameters of the numerical model—for instance, wavespeed, pipe and fluid properties, initial and boundary conditions, etc. In the same way, the uncertainties in the pressures transducers, such as operating mode, environment, calibration frequency, among others, had an influence on the results.

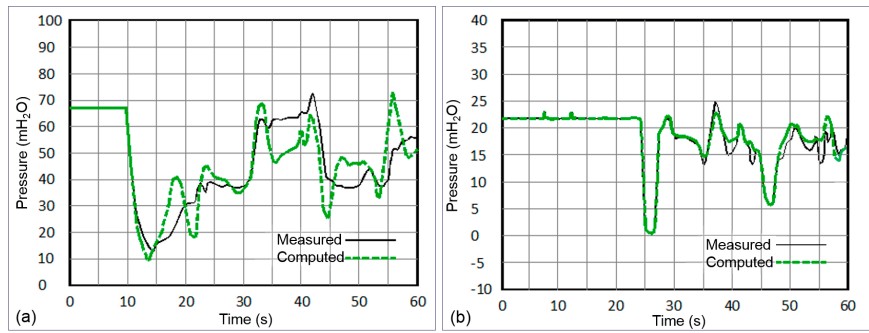

**Figure 13.** Measured and computed pressure traces (**a**) at discharge header of Pumping Plant 1 and (**b**) at the air valve located at station 13 + 600.

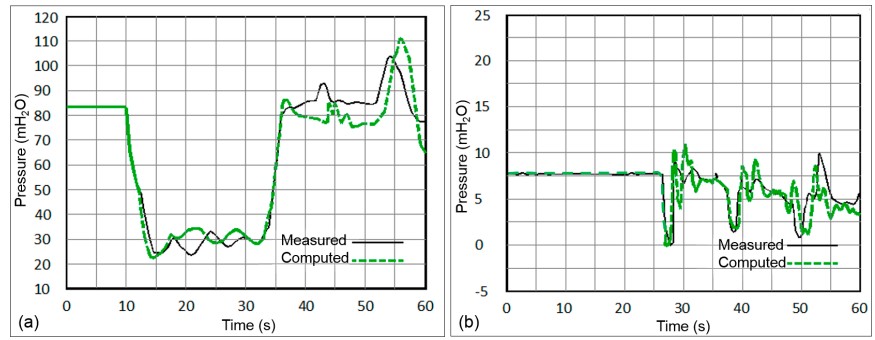

**Figure 14.** Measured and computed pressure traces (**a**) at discharge header of Pumping Plant 2 and (**b**) at the air valve located at station 38 + 800.

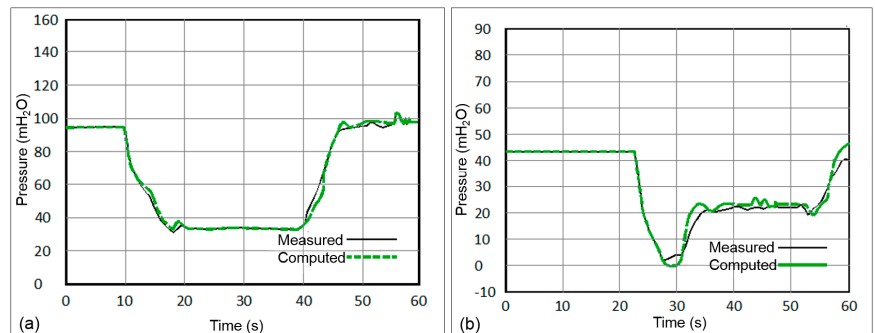

**Figure 15.** Measured and computed pressure traces (**a**) at discharge header of Pumping Plant 3 and (**b**) at the air valve located at station 54 + 560.

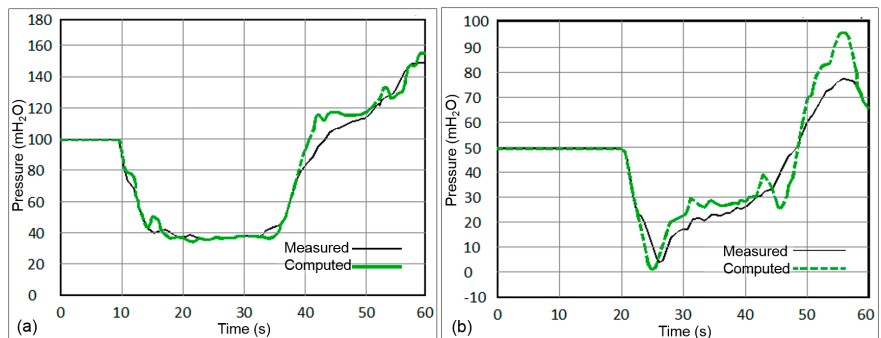

**Figure 16.** Measured and computed pressure traces (**a**) at discharge header of Pumping Plant 4 and (**b**) at the air valve located at station 73 + 476.

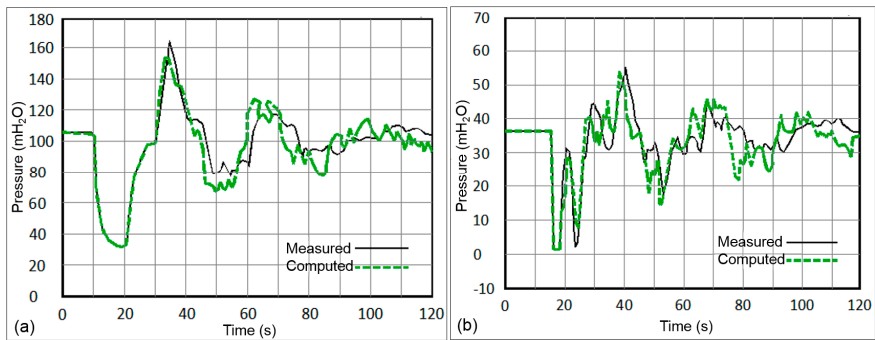

**Figure 17.** Measured and computed pressure traces (**a**) at discharge header of Pumping Plant 5 and (**b**) at the air valve located at station 84+447.

## 6. Conclusions

This paper presents the solution in order to ensure that a pumping pipeline of large dimensions operates properly to avoid potential damage due to the occurrence of hydraulic transients. The original design of the aforementioned pipeline was not suitable because the devices for transient control were undersized and insufficient to suppress downsurge pressures when the simultaneous failure of the five pumps occurs at the five pumping stations caused by power cut, as a result of the moment of inertia of all the pumps being incorrectly considered during the design. Furthermore, the hydraulic transient analysis revealed that the sub-atmospheric pressures at the first, third and fifth pipeline segments may be reduced to the vapor pressure of the liquid, which could lead to water column separation. Hence, supplementary surge suppression devices were needed to prevent potential catastrophic damage to the pipeline.

With the intention of effectively protecting the pumping pipeline, an exhaustive transient analysis was conducted for the design and testing of additional devices for transient control, considering the simultaneous power failure of all the pumps at the five pumping stations. The results revealed that the best option to uplift the minimum pressures and to reduce the maximum pressures in the system to within acceptable limits was to install three air chambers at the discharge header of each pumping station to supplement the original system for surge control. It is important to highlight that the five pumps of each pumping station were never put into operation until the system for surge control was improved.

Due to the manufacturing process and installation of the air chambers at each discharge header taking some time, and owing to the necessity to start supplying water to the population, the authors were prompted to find a cost-effective solution that could be easily and rapidly implemented. An unconventional form of protecting the pipeline against downsurge pressures was recommended and this allowed for delivering a water discharge of about 2.4 $m^3$/s. Results showed that the most efficient measure to protect the pipeline against downsurge pressures was to operate two of the five units per plant with the original control devices and install additional air valves at strategic locations to permit large volumes of air to come in the pipeline through air valves, when the hydraulic grade line falls below its elevation once the two pumps fail at each station. It is well known that a manner to diminish the transient pressures in pipelines is by means of large air pockets, provided that they remain stationary in a suitable position, normally at high points of pipelines. It has been demonstrated that the effect of large pockets on transient pressures can be beneficial, since they perform as air cushions suppressing the energy of transient pressure waves, replicating the behavior of air chambers. Moreover, it is worth noting that it is not a common practice to introduce large volumes of air into pipelines of large dimensions to evade negative pressures. Likewise, it is important to highlight that to the authors' knowledge of this investigation presents one of the first applications of large air pockets to avoid negative pressures in a large pumping pipeline system. Furthermore, it is imperative to emphasize that, once air is admitted into the pipeline, caution must be exercised for expelling air while refilling

the pipeline. The air should be vented slowly from the pipeline, since the entrapped air pockets of a certain size may generate very high pressures.

Furthermore, once the selected air valves were installed, field measurements were performed. They consisted of recording transient pressures following the simultaneous failure of two pumps at each pumping station. Subsequently, the surge modelling results were compared with the data registered during field pressure measurements. It was noticed that the simulated and measured pressures are in good agreement. Therefore, it can be stated that the large air pockets in combination with an existing system for transient control adequately protect the pumping system, avoiding negative pressures and potential damage to the pipeline.

**Author Contributions:** Conceptualization, R.B.C.-P. and O.P.-E.; Methodology, R.B.C.-P., O.P.-E. and L.G.C.P.; Software, R.B.C.-P. and L.G.C.P.; Validation, L.G.C.P. and A.S.H, E.A.R.-C. and G.J.C.-P.; Formal Analysis, R.B.C.-P., O.P.-E., L.G.C.P. and A.S.H; Investigation, R.B.C.-P., A.S.-H., E.A.R.-C. and G.J.C.-P.; Resources, R.B.C.-P. and A.S.-H.; Data Curation, R.B.C.-P., L.G.C.P. and A.S.H; Writing—Original Draft Preparation, O.P.-E. and R.B.C.-P.; Writing—Review and Editing, O.P.-E. and R.B.C.-P.; Supervision, R.B.C.-P.; Project Administration, R.B.C.-P.

**Funding:** This research received no external funding.

**Conflicts of Interest:** The authors declare no conflict of interest.

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
