# Peer review of "Protecting a Pumping Pipeline System from Low Pressure Transients by Using Air Pockets: A Case Study"

_water, doi:10.3390/w11091786_

Round 1
Reviewer 1 Report
This paper presents an interesting study of the effect of air valve to regulate the minimum and maximum attained pressures in a real pipeline. A very good case study has been selected. Some details are missed about the case study as requested in following comments. The methodology and the structure of the paper is organised very well. Some descriptions are repeated in some parts of the paper which reduces the quality of the paper in terms of reading convenience. The results are provided with enough graphs. However, some explanations about used terms are needed as stated in the comments. There is a lack of discussions in temporal analysis of the results to see what is the time difference between the pump shut down and occurring the major events. As authors stated, this real model suffers from some initial designing problems. Maybe the main reason for considerable change in results is due to this fact. It needs to be more detailed. It would be useful if authors could show how feasible can be this type of analysis for other systems. In general, the paper requires minor revision based on following comments.
Line 129: It would be interesting if you could show some comparison between different scenarios.
Line 134 to 137: This information was repeated several times in different sections. It would be better if you avoid this repetition.
Line 144: This kind of classifications are a bit tricky because nowadays manufacturers produce different range of valves which mostly are multi-functional. For example, there is another type named vacuum breaker valve. So, I suggest to avoid this classification.
Line 149: What is reference 27? It is not written correctly.
Line 185: please rephrase as “Moreover, if the head losses at the air valve and junction are neglected, then …”
Line 210: I suggest using p* for absolute pressure
Line 243: please rephrase as “Since various researchers state that the most serious situation in a pumping station occurred when all the pumping devices fail 245 simultaneously caused by power failure [3 ,5,24], this situation was first …”
Line 260: Please consider rephrasing this sentence. It is not clear.
Line 281: I think you mean ref, no. 31. Please, in general, reconsider the numbering of the references in the manuscript.
Line 292: please correct as “… advanced devices that combine the air release and vacuum …”
Line 322: What is the star (*) sign for in Table 4. Please describe it.
Line 368: Please give some details about the reservoirs in the pump stations.
Line 368: I assume that the original heads are those calculated by the manufacturer and Real ones are those calculated with real moment of inertia. Please confirm and describe this better.
Line 377: In some rare cases in Figure 5 and Figure 6 the maximum head with additional air chamber is higher than other cases. How authors describe this?
Line 393: Please provide some details on the formed pockets that remained stationary at valve location from practical point of view.
Line 430: Please add some discussions to this section in terms of temporal occurrence of pressure fluctuations considering the time of the pump shut down.
Line 487: please remove “The necessity to start supplying water to the population as soon as possible, whereas the air chambers are manufactured and installed.”
Author Response
We thank the reviewer for reading the manuscript carefully and for providing us constructive comments and suggestions which we find very helpful. These comments and suggestions will improve the readability of the manuscript and clarify the aim of this study.
We have read carefully the comments, which include technical recommendations and we have revised the paper to address all the recommendations as requested.
Reply to Reviewer 1’s comments/suggestions
Q1. This paper presents an interesting study of the effect of air valve to regulate the minimum and maximum attained pressures in a real pipeline. A very good case study has been selected. Some details are missed about the case study as requested in following comments. The methodology and the structure of the paper is organised very well. Some descriptions are repeated in some parts of the paper which reduces the quality of the paper in terms of reading convenience. The results are provided with enough graphs. However, some explanations about used terms are needed as stated in the comments. There is a lack of discussions in temporal analysis of the results to see what is the time difference between the pump shut down and occurring the major events. As authors stated, this real model suffers from some initial designing problems. Maybe the main reason for considerable change in results is due to this fact. It needs to be more detailed. It would be useful if authors could show how feasible can be this type of analysis for other systems. In general, the paper requires minor revision based on following comments.
A1. Following the recommendations, the manuscript was revised and some paragraphs of the paper were rewritten to improve the readability of the manuscript.
Q2. Line 129: It would be interesting if you could show some comparison between different scenarios.
A2. The comparison between different scenarios is shown in figures 3 to 7. These figures present the transient pressures caused by the simultaneous failure of 5 pumps, considering the incorrect moment of inertia of the pumps used during the design stage, the correct moment of inertia provided by the pump manufacturer, as well as the correct moment of inertia and the three additional air chambers installed at the discharge header of each pumping station.
Q3. Line 134 to 137: This information was repeated several times in different sections. It would be better if you avoid this repetition.
A3. In the revised manuscript, we avoid the repetition of this information.
Q4. Line 144: This kind of classifications are a bit tricky because nowadays manufacturers produce different range of valves which mostly are multi-functional. For example, there is another type named vacuum breaker valve. So, I suggest to avoid this classification.
A4. In the revised manuscript, we removed this classification.
Q5. Line 149: What is reference 27? It is not written correctly.
A5. In the revised manuscript, we wrote correctly all references.
Q6. Line 185: please rephrase as “Moreover, if the head losses at the air valve and junction are neglected, then …”
A6. The paragraph was rewritten to improve the readability of the manuscript.
Q7. Line 210: I suggest using p* for absolute pressure
A7. The symbol p* is used for absolute pressure in the revised version of the manuscript.
Q8. Line 243: please rephrase as “Since various researchers state that the most serious situation in a pumping station occurred when all the pumping devices fail 245 simultaneously caused by power failure [3 ,5,24], this situation was first …”
A8. The paragraph was rewritten to improve the readability of the manuscript.
Q9. Line 260: Please consider rephrasing this sentence. It is not clear.
A9. Following this suggestion, the paragraph was revised and rewritten to improve the readability of the manuscript.
Q10. Line 281: I think you mean ref, no. 31. Please, in general, reconsider the numbering of the references in the manuscript.
A10. In the revised manuscript the numbering of the references was corrected.
Q11. Line 292: please correct as “… advanced devices that combine the air release and vacuum …”
A11. In revised manuscript, we wrote combine instead of combines.
Q12. Line 322: What is the star (*) sign for in Table 4. Please describe it.
A12. The star (*) denotes the new air valves installed in the pipeline.
Q13. Line 368: Please give some details about the reservoirs in the pump stations.
A13. In the revised version of the manuscript we give details about the suction reservoirs.
Q14. Line 368: I assume that the original heads are those calculated by the manufacturer and Real ones are those calculated with real moment of inertia. Please confirm and describe this better.
A14. The original heads (heads obtained in the design) are those calculated with the incorrect moment of inertia, while the real ones are those evaluated with the correct moment of inertia. Likewise, the explanation related with the moment of inertia and the results was improved.
Q15. Line 377: In some rare cases in Figure 5 and Figure 6 the maximum head with additional air chamber is higher than other cases. How authors describe this?
A15. In the original manuscript we plotted the maximum head envelopes considering smaller air chambers. In the revised manuscript the correct maximum head envelopes are presented.
Q16. Line 393: Please provide some details on the formed pockets that remained stationary at valve location from practical point of view.
A16. In the revised version of the manuscript, the description of the air pockets was improved.
Q17. Line 430: Please add some discussions to this section in terms of temporal occurrence of pressure fluctuations considering the time of the pump shut down.
A17. In the revised version of the manuscript, this paragraph was improved.
Q18. Line 487: please remove “The necessity to start supplying water to the population as soon as possible, whereas the air chambers are manufactured and installed.”
A18. We removed these two lines.
Reviewer 2 Report
This manuscript is, by its title, intended as a case study and the specific application doesn’t seem to be very well suited for that purpose. In the discussion, it is implied that the solution is unconventional and I would like to know more about that. After all, the reason that these types of air valves are used is in this type of application. The discussion represents this as a temporary solution and I would like to know more about why that should be the case. The real lesson to this application should be not to analyze a system using incorrect data on system elements and don’t wait until the project is about to be implemented to discover this. Otherwise, the approach described in the manuscript is not required; ideally this is not a common situation to be encountered. It would be of considerable interest to know more about the power supply for the system. Is it constructed so that a simultaneous failure at pumping stations spread over 90 km due to the interconnections in the electrical grid even likely?
There is another major issue that seems to underly the manuscript and at least requires additional discussion. I believe that the authors understand that merely experiencing sub-atmospheric pressures doesn’t lead to water column separation but that the pressures need to be below the vapor pressure of the flowing liquid. For example, it isn’t true that, as stated in line 350, most of the fifth pipeline segment would experience water column separation but a more accurate statement is that much of that segment could experience sub-atmospheric pressure and one location near 82000 feet seems susceptible to water column separation (and possibly at the end of the pipe segment depending on local geometry which is unclear). It seems as though a different design objective is being applied than what is being discussed.
Some more specific comments
Lines 141-143. This sentence is totally vague. I have no idea what is being intended
The methodology in section 3 is not novel so it is unclear that the detail here is even needed since it is available in standard references.
There is reference in several locations of the need that the air pockets be stationary and not transported with the flow, but it seems that this is an implied assumption. I have no doubt that this is the case in this application, although the exact location of the valves might be important, but it seems inappropriate not to even discuss this issue.
In Table 4, what do the * represent? I am guessing that it has to do with initially designed or newly added valves, but it is not explained.
Line 334 “whether” should be “when”
Line 341 not sure what that discussion about heightening means
A little more discussion about how the system is supposed to function would be in order. Why can you only use two pumps at each station? I assume this has something to do with increasing demand over time, but this is relevant to whether this case study is applicable to other problems.
Line 440 First sentence is meaningless statement unless some more context is supplied? What sort of uncertainty in the measurements?
Several locations in the manuscript where subject and verb are inconsistent (singular or plural, tense)
Author Response
We thank the reviewer for reading the manuscript carefully and for providing us constructive comments and suggestions which we find very helpful. These comments and suggestions will improve the readability of the manuscript and clarify the aim of this study.
We have read carefully the comments, which include technical recommendations and we have revised the paper to address all the recommendations as requested.
Reply to Reviewer 2’s comments/suggestions
Q1. This manuscript is, by its title, intended as a case study and the specific application doesn’t seem to be very well suited for that purpose. In the discussion, it is implied that the solution is unconventional and I would like to know more about that. After all, the reason that these types of air valves are used is in this type of application.
A1. Following the recommendations, the manuscript was revised and some paragraphs of the paper were rewritten to improve the readability of the manuscript.
The authors agree with the reviewer that air valves are used for decreasing the potential negative pressure within the pipeline during a draining operation, pipeline rupture, or pump failure. However, it is important to highlight that it is not a common practice to admit large quantities of air into pipelines of large dimensions to avoid negative pressures.
Q2. The discussion represents this as a temporary solution and I would like to know more about why that should be the case. The real lesson to this application should be not to analyze a system using incorrect data on system elements and don’t wait until the project is about to be implemented to discover this. Otherwise, the approach described in the manuscript is not required; ideally this is not a common situation to be encountered. It would be of considerable interest to know more about the power supply for the system. Is it constructed so that a simultaneous failure at pumping stations spread over 90 km due to the interconnections in the electrical grid even likely?
A2. It is important to highlight that the authors of this paper were not involve in the design of the system under study. Likewise, the five pumps of each plant were never put into operation until the system for transient control was improved. We request the information of the power supply for the system, but because of the bureaucracy it was practically impossible to get it.
Since the manufacturing and installation of the three air chambers at each of the pumping plants would take some time, and due to the necessity to start supplying water to the population as soon as possible, prompt the authors to find a cost-effective solution that could be easily and rapidly implemented, but also offering an adequate protection to the pipeline against transient pressures. With this purpose, preliminary numerical simulation were developed. The system was analyzed for the sudden power failure to the pump stations with one, two, three and four pumps in operation with its correct moment of inertia. In all instances, the results indicated that the minimum head enveloped intersects some sections of the pipeline profile, producing negative pressures. Therefore, it can be stated that the original system for transient control, as well as the existing air valves are unable of mitigating transient pressures caused by power failure, no matter the number of pumps in operation.
Based on the above, the authors suggested an unconventional form of protecting the pipeline against low pressures to put the system into operation with less than five pumping units. The temporary solution proposed was to allow large quantities of air to enter the pipeline through air valves, when the hydraulic grade line falls below its elevation. When this occurs, large air pockets remain stationary at high points of the line, because the buoyancy of the pockets prevent them from being dragged downstream by the drag force of water flow. Likewise, it is well known that the effect of large pockets on transient pressures may be beneficial. Further, various researchers have demonstrated that large air pocket can act as an effective accumulator suppressing the energy of pressure waves, replicating the behavior of an air chamber.
Subsequently numerous hydraulic transient simulations caused by power failure to pumps were conducted to select and size more appropriate air valves in order to enter the necessary volume of air to the pipeline to ensure that large air pockets remain stationary in adequate locations. Results show the best solution is to operate 2 of the 5 units per plant with the original control devices and install additional air valves at strategic locations to prevent the occurrence of negative pressures. Likewise, it is important to highlight that it is not a common practice to admit large quantities of air into pipelines of large dimensions to avoid negative pressures.
Q3. There is another major issue that seems to underly the manuscript and at least requires additional discussion. I believe that the authors understand that merely experiencing sub-atmospheric pressures doesn’t lead to water column separation but that the pressures need to be below the vapor pressure of the flowing liquid. For example, it isn’t true that, as stated in line 350, most of the fifth pipeline segment would experience water column separation but a more accurate statement is that much of that segment could experience sub-atmospheric pressure and one location near 82000 feet seems susceptible to water column separation (and possibly at the end of the pipe segment depending on local geometry which is unclear). It seems as though a different design objective is being applied than what is being discussed.
A3. Agree, the paragraphs related with the pressures obtained during the simulations were rewritten to improve the readability of the manuscript.
Q4. Lines 141-143. This sentence is totally vague. I have no idea what is being intended
A4. The paragraph was rewritten to improve the readability of the manuscript.
Q5. The methodology in section 3 is not novel so it is unclear that the detail here is even needed since it is available in standard references.
A5. No doubt there is a wealth of literature addressing the problem of hydraulic transients. However, standard literature on fluid transients do not provide a rigorous analysis on the analysis of the effect of air pocket volumes introduced into the investigated pumping pipeline via the air valves. Further, one aim of the paper is to present a methodology to evaluate fluid transients with housing entrapped air pockets in pumping pipelines. Therefore, we consider all equations are needed in the text. This will provide non-experts in this field with a summarized algorithm, which can be easily followed to solve a similar problem.
Q6. There is reference in several locations of the need that the air pockets be stationary and not transported with the flow, but it seems that this is an implied assumption. I have no doubt that this is the case in this application, although the exact location of the valves might be important, but it seems inappropriate not to even discuss this issue.
A6. In the revised version of the manuscript, we gave more details about the behavior of large air pockets.
Q7. In Table 4, what do the * represent? I am guessing that it has to do with initially designed or newly added valves, but it is not explained.
A7. The star (*) denotes the new air valves installed in the pipeline.
Q8. Line 334 “whether” should be “when”
A8. In revised manuscript, we wrote when instead of whether.
Q9. Line 341 not sure what that discussion about heightening means
A9. In the revised manuscript, we wrote increase instead of heightening.
Q10. A little more discussion about how the system is supposed to function would be in order. Why can you only use two pumps at each station? I assume this has something to do with increasing demand over time, but this is relevant to whether this case study is applicable to other problems.
A10. In the revised manuscript, we gave more details about how the system is supposed to function, and why we use two pumps at each station to start supplying water to the population.
Q11. Line 440 First sentence is meaningless statement unless some more context is supplied? What sort of uncertainty in the measurements?
A11. The paragraphs of this section were rewritten to improve the readability of the manuscript.
Q12. Several locations in the manuscript where subject and verb are inconsistent (singular or plural, tense)
A12. Following the recommendations, the manuscript was revised and some paragraphs of the paper were rewritten to improve the readability of the manuscript.
Round 2
Reviewer 2 Report
The manuscript has been improved since the first version, but some residual problems still remain which should be capable of being corrected. There are still quite a lot of grammar errors in the text, probably more now than in the earlier version because of the addition of new material. One issue that should be clear to a careful reader is that in several places, for example the headings to sections 5.1 and 5.2 do not indicate that the five and two pumps are per pump station; this issue is addressed properly in other places in the manuscript. The other point that I made in the previous review related to negative pressures versus water column separation was acknowledged in the response, but not really well addressed in the manuscript. The numerical simulation, if trustworthy, is capable of predicting whether column separation occurs or not. In the text, it is now mentioned that negative pressures occur in the predictions that could lead to column separation. If I understand the design objective, it is to avoid sub-atmospheric pressures which is a different criterion than avoiding column separation. Why don't you just clearly state that as the design objective and it avoids confusion?
Author Response
We thank the reviewer for reading the manuscript carefully and for providing us constructive comments and suggestions which we find very helpful. These comments and suggestions will improve the readability of the manuscript and clarify the aim of this study.
We have read carefully the comments, which include technical recommendations and we have revised the paper to address all the recommendations as requested.
Q1. The manuscript has been improved since the first version, but some residual problems still remain which should be capable of being corrected. There are still quite a lot of grammar errors in the text, probably more now than in the earlier version because of the addition of new material.
A1. Following the recommendations, the manuscript was revised and some paragraphs of the paper were rewritten to improve the readability of the manuscript.
Q2. One issue that should be clear to a careful reader is that in several places, for example the headings to sections 5.1 and 5.2 do not indicate that the five and two pumps are per pump station; this issue is addressed properly in other places in the manuscript.
A2. In the new version of the manuscript, we indicated that the five and two pumps are per pump station.
Q3.The other point that I made in the previous review related to negative pressures versus water column separation was acknowledged in the response, but not really well addressed in the manuscript. The numerical simulation, if trustworthy, is capable of predicting whether column separation occurs or not. In the text, it is now mentioned that negative pressures occur in the predictions that could lead to column separation. If I understand the design objective, it is to avoid sub-atmospheric pressures which is a different criterion than avoiding column separation. Why don't you just clearly state that as the design objective and it avoids confusion?
A3. We improved the explanation related with the water-column separation.